# Mapping of a Major QTL, *qBK1^Z^*, for Bakanae Disease Resistance in Rice

**DOI:** 10.3390/plants10030434

**Published:** 2021-02-25

**Authors:** Sais-Beul Lee, Namgyu Kim, Sumin Jo, Yeon-Jae Hur, Ji-Youn Lee, Jun-Hyeon Cho, Jong-Hee Lee, Ju-Won Kang, You-Chun Song, Maurene Bombay, Sung-Ryul Kim, Jungkwan Lee, Young-Su Seo, Jong-Min Ko, Dong-Soo Park

**Affiliations:** 1National Institute of Crop Science, Milyang 50424, Korea; pappler@korea.kr (S.-B.L.); tnals88319@korea.kr (S.J.); gjduswodi@naver.com (Y.-J.H.); minitia@korea.kr (J.-Y.L.); hy4779@korea.kr (J.-H.C.); ccriljh@korea.kr (J.-H.L.); kangjw81@korea.kr (J.-W.K.); songyc@korea.kr (Y.-C.S.); kojmin@korea.kr (J.-M.K.); 2Department of Microbiology, Pusan National University, Pusan 46241, Korea; titanic622@pusan.ac.kr (N.K.); yseo2011@pusan.ac.kr (Y.-S.S.); 3International Rice Research Institute, Pili Drive, Los Baños 4031, Laguna, Philippines; m.bombay@irri.org (M.B.); s.r.kim@irri.org (S.-R.K.); 4College of Natural Resources and Life Science, Dong-A University, Pusan 49135, Korea; jungle@dau.ac.kr

**Keywords:** *Oryza sativa* L., bakanae disease, *Gibberella fujikuroi*, QTL mapping—*qBK1^Z^*

## Abstract

Bakanae disease is a fungal disease of rice (*Oryza sativa* L.) caused by the pathogen *Gibberella fujikuroi* (also known as *Fusarium fujikuroi*). This study was carried out to identify novel quantitative trait loci (QTLs) from an *indica* variety Zenith. We performed a QTL mapping using 180 F_2:9_ recombinant inbred lines (RILs) derived from a cross between the resistant variety, Zenith, and the susceptible variety, Ilpum. A primary QTL study using the genotypes and phenotypes of the RILs indicated that the locus *qBK1^z^* conferring bakanae disease resistance from the Zenith was located in a 2.8 Mb region bordered by the two RM (Rice Microsatellite) markers, RM1331 and RM3530 on chromosome 1. The log of odds (LOD) score of *qBK1^z^* was 13.43, accounting for 30.9% of the total phenotypic variation. A finer localization of *qBK1^z^* was delimited at an approximate 730 kb interval in the physical map between Chr01_1435908 (1.43 Mbp) and RM10116 (2.16 Mbp). Introducing *qBK1^z^* or pyramiding with other previously identified QTLs could provide effective genetic control of bakanae disease in rice.

## 1. Introduction

Bakanae disease, which means foolish seedling in Japanese, was firstly identified in 1828 in Japan [1], and is widely distributed in temperate zones as well as tropical environments and occurs throughout rice growing regions of the world [2].

Four *Fusarium* species including *F. andiyazi*, *F. fujikuroi*, *F. proliferatum* and *F. verticillioides* in the *G. fujikuroi* species complex have been associated with bakanae disease in rice [3]. This disease is typically a seed-borne fungus, but may occur when the pathogen is present in plant material or soil. Infected seeds/plants result in secondary infections [4], which spread through wind or water. Bakanae disease has different symptoms such as tall, lanky tillers with pale green flag leaves. Infected plants also have fewer tillers, and plants surviving till maturity bear only empty panicles [5], resulting in yield loss [6,7]. Low plant survival and high spikelet sterility [5] may account for yield losses of up to 50% in Japan [7], 3.0–95% in India [8,9,10], 3.7–14.7% in Thailand, 5–23% in Spain, 40% in Nepal [10], 6.7–58.0% in Pakistan [11], 75% in Iran [12] and to 28.8% in Korea [13]. Germinating rice seeds in seed boxes for mechanical transplantation has caused many problems associated with diseases [14] including bakanae disease, which are not considered serious in direct seeding. Hot water immersion and fungicide treatment are the most common ways of seed disinfection [10,15,16]. However, both the hot water treatment and application of fungicide are insufficient to control bakanae disease. Thermal effect does not reach the pericarp of the severely infected rice seeds. The application of fungicides is not functioning well for destroying the spores of this fungal pathogen, and some pathogen showed resistance to the fungicides [13,17,18,19]. Therefore, the genetic improvement of rice using the quantitative trait loci (QTLs)/genes providing bakanae disease resistance would be a more effective way to control bakanae disease.

Several QTLs associated with bakanae disease resistance have been identified and those can be used for marker-assisted selection in rice breeding as well as for understanding the mechanisms of resistance. Yang et al. [20] identified two QTLs located on chromosome 1 and chromosome 10 by in vitro evaluation of the Chunjiang 06/TN1 doubled haploid population. Hur et al. [21] identified a major QTL, *qBK1*, on chromosome 1 from 168 BC_6_F_4_ near isogenic lines (NILS) generated by crossing the resistant *indica* variety Shingwang with susceptible *japonica* variety Ilpum. Lee et al. [22] delimited the location of *qBK1* to 35 kb interval between two InDel (Insertion–deletion) markers, InDel 18 (23.637 Mbp) and InDel 19-14 (23.672 Mbp). Fiyaz et al. [23] identified three QTLs (*qBK1.1*, *qBK1.2*, and *qBK1.3*) on chromosome 1 and one QTL (*qBK3.1*) on chromosome 3 from the highly resistant variety Pusa 1342. Ji et al. [24] mapped a major QTL (*qFfR1*) in 22.56–24.10 Mbp region on chromosome 1 from a resistant Korean *japonica* variety Nampyeong. Volante et al. [25] identified *qBK1_628091* (0.6–1.0 Mbp on chromosome 1) and *qBK4_31750955* (31.1–31.7 Mbp on chromosome 4) by GWAS (genome-wide association study) approach using 138 *japonica* rice germplasms. Kang et al. [26] discovered the QTL *qFfR9* at 30.1 centimorgan (cM) on chromosome 9 from a *japonica* variety Samgwang. Lee et al. [16] found the QTL *qBK1^WD^* located between markers chr01_13542347 (13.54 Mb) and chr01_15132528 (15.13 Mb) from the *japonica* variety Wonseadaesoo. They also found that resistance of gene pyramided lines harboring two QTLs, *qBK1^WD^* and *qBK1*, was significantly higher than those with only *qBK1^WD^* or *qBK1.* Identifying new resistance genes from diverse sources is important for rice breeding programs to acquire durable resistance against bakanae disease by either enhancing the resistance level, helping to overcome the breakdown of resistance genes, or both. In this study, we aimed to provide a new genetic source, *qBK1^z^* with detailed gene locus information for developing resistant rice lines which contains single or multiple major QTLs to enhance bakanae disease resistance.

## 2. Results

### 2.1. Bakanae Disease Bioassay in Parents and F_2:9_ RILs

The proportion of healthy Zenith (resistant) and Ilpum (susceptible) plants was evaluated with 10 biological replicates after inoculation of the virulent *F. fujikuroi* isolate CF283 [27]. Most Zenith plants did not exhibit a thin and yellowish-green phenotype, which is a typical symptom of bakanae disease, unlike Ilpum (Figure 1A).

The proportion of healthy Zenith plants was 63.2 ± 11.8% (ranging from 42.7% to 79.3%), which was significantly different from that of Ilpum (14.3 ± 11.4%, ranging from 3.3% to 37.0%) (Figure 1B). Zenith and Ilpum were further inoculated with green fluorescent protein (GFP)-tagged *F. fujikuroi* isolate CF283. Ten days after inoculation, plants with typical disease symptom of each variety were subjected to a confocal microscopy analysis. Confocal imaging of radial and longitudinal sections of the basal stem showed that the fungus penetrated and was localized easily and abundantly at vascular bundle, mesophyll tissue and hypodermis in the susceptible Ilpum variety, while only a background level of GPF signal was detected in the resistant Zenith (Figure 2).

### 2.2. QTL Analysis and Mapping of qBK1^z^ Using 180 F_2:9_ RILs

Based on the bakanae disease bioassay (proportion of healthy plants), the 180 F_2:9_ RIL population exhibited continuous distribution (ranged from 0% to 98.0%; Figure 3), which quantitatively confirmed the inheritance of bakanae disease resistance.

We selected 164 markers showing polymorphism between Ilpum and Zenith from 1150 RM markers (http://gramene.org, accessed on 12 August 2018) tested. which covering the whole rice chromosome (Figure A1). The genetic linkage map of Ilpum and Zenith for primary mapping was constructed with 164 polymorphic markers covering a total length of 3140 cM with average interval of 19.14 cM as described by Lee et al. [28]. Primary QTL mapping using the 180 F_2:9_ populations showed that a significant QTL associated with bakanae disease resistance at the seedling stage was located between the SSR markers, RM1331 and RM3530 on chromosome 1, and it was designated *qBK1^z^.* The LOD score of *qBK1^z^* was 13.43, which accounted for 30.9% of the total phenotypic variation (Table 1).

A finer localization of *qBK1^z^* was determined by analyzing the chromosome segment introgression lines in the region detected from primary mapping. The *qBK1^z^* region between RM1331 and RM3530 from primary mapping was narrowed downed with an additional 55 SSR markers and 12 InDel markers designed for the insertion/deletion sites based on the differences between the *japonica* (http://www.gramene.org, accessed on 12 August 2018) and *indica* (http://rice.genomics.org.cn, accessed on 12 August 2018) sequences. Four SSR markers and six InDel markers were selected as polymorphic markers between the parents to narrow down the position of the *qBK1^z^* region (Table A1). Finally, seven homozygous recombinants were selected from the F_2:9_ lines using 14 markers in the 2.8 Mb region around the SSR markers RM1331 and RM3530 (Figure 4 and Figure 5).

The proportion of healthy plants of the seven homozygous recombinants was evaluated with three biological replicates according to Duncan’s new multiple range test. Based on this bioassay, lines classified to Group “a” were regarded as resistant, and Group “b” as susceptible (Figure 4). Considering the genotype and the phenotype of the recombinants, it is clear that *qBK1^Z^* conferring resistance to bakanae was an approximate 730 kb interval delimited by the physical map between Chr01_1435908 (1.43 Mbp) and RM10116 (2.16 Mbp).

## 3. Discussion

Rice varieties with a single resistance gene are at an increased risk of being overcome by new pathological races [16,28]. The development of a rice variety with a higher level of resistance against bakanae disease is a major challenge in many countries [21,23,29,30,31]. In this study, we identified *qBK1^z^* locus related to bakanae disease resistance based on genotype and phenotype analyses of homozygous recombinants on the recombinant progeny of Ilpum and Zenith, using SSR and newly developed InDel markers.

It was reported that successful infection of *Fusarium* species is a complex process that includes adhesion, penetration (through wounds, seeds, stomatal pores) and subsequent colonization within and between cells [32,33]. Lee et al. [16] revealed that the fungus *F. fujikuroi* was more abundant in the stem of the susceptible variety than it was in the resistant one. Elshafey et al. [34] indicated that *F. fujikuroi* prefer to grow in aerenchyma, pith, cortex and vascular bundle of both sheath and stem of rice. In this study, we examined both the localization and abundance of *F. fujikuroi* isolate CF283 in the basal stem of rice using GFP-tagged *F. fujikuroi* isolate CF283 (Figure 2). Consistent with previous reports [16,34], *F. fujikuroi* isolate CF283 was extensively observed on the vascular bundle, mesophyll tissue and hypodermis of infected stems in susceptible the Ilpum variety, whereas this was rarely observed in resistant Zenith.

Many QTLs on bakanae disease resistance have been identified on chromosome 1. Three QTLs, *qBK1^z^*, *qBK1.2* and *qBK1.3*, were found in a similar region in spite of the different source of resistant varieties (Figure 6).

Fiyaz et al. [23] mapped *qBK1.1* to a 20 kb region between markers RM9 and RM11232 from the Pusa 1121/Pusa1342 cross. These authors hypothesized that *qBK1.1* and *qBK1* [21] might be the same QTL as they had overlapping positions. Ji et al. [24] found that QTL *qFfR1* was located in a 230 kb region of rice chromosome 1 in Korean *japonica* variety Nampyeong, and suggested that the three QTLs *qBK1, qBK1.1* and *qFfR1* might indicate the same gene. Lee et al. [22] narrowed down the position of the *qBK1* locus to a 35 kb region between InDel 18 and InDel 19-14, and revealed that location of *qBK1* is close to those of *qBK1.1* and *qFfR1*, and do not overlap each other. Two additional QTLs including *qB1* from Chunjiang 06 [20] and *qBK1^WD^* from Wonseadaesoo [16] were also found on chromosome 1. Gene pyramiding via phenotypic screening assays for crop breeding is considered to be difficult and often impossible due to dominance and epistatic effects of genes governing disease resistance, and the limitation of screenings being all year-round [35]. Pyramiding of multiple resistant QTLs/genes by using marker-assisted breeding (MAB) in a single plant might confer either higher, durable, or both, resistances against bakanae disease. The effects of pyramiding resistance genes have been observed for several plant-microbe interactions. Pyramiding three bacterial blight resistance genes resulted in a high level of resistance and were expected to provide a durable pathogen resistance [36,37]. On the other hand, pyramiding of resistant genes resulted in a level of resistance that was comparable to or even lower to than that of the line with a single gene. For example, Yasuda et al. [38] reported rice lines with pairs of blast resistance genes to be only comparable to lines with a single gene which may have a stronger suppressive effect. Our previous study of bakanae disease resistance [16] revealed that the gene pyramided lines harboring *qBK1^WD^* + *qBK1* had a much higher levels of resistance than those possessing either *qBK1^WD^* or *qBK1*. The novel QTL, *qBK1^Z^*, identified in this study can be utilize in MAB and gene pyramiding to achieve higher resistance in many bakanae disease prone rice growing areas.

In this study, we identified a new major QTL *qBK1^z^* conferring bakanae disease resistance from a new genetic source of *indica* variety, Zenith. Through QTL analysis and fine mapping, we narrowed down the *qBK1^z^* locus into 730 kb on the short arm of chromosome 1 which is a novel locus compared with all the previously identified bakanae disease resistance QTLs. Together with the previously identified QTLs of the bakanae disease resistance, the new *qBK1^z^* can be introduced to the elite favorable background varieties by a marker-assisted backcrossed breeding. Furthermore, the information of the localization and different abundance on the vascular bundle, mesophyll tissue and hypodermis of infected stems of infected rice between resistant and susceptible varieties will be useful for further studying an interaction between the pathogen (*F. fujikuroi*) and rice host plants.

## 4. Materials and Methods

### 4.1. Plant Materials

Zenith, a medium grain type *indica* variety from USA released in 1936, was identified as resistant to bakanae disease in a preliminary screening of rice germplasm (data not shown). We generated 180 F_2:9_ RILs from a cross between susceptible variety, Ilpum, and resistant variety, Zenith, for QTL analysis. The population was developed in the experimental fields of the National Institute of Crop Science in Miryang, Korea.

### 4.2. Evaluation of Bakanae Resistance and Statistical Analysis

The inoculation and evaluation of bakanae disease were conducted using a method described by Lee et al. [22]. The isolate CF283 of *F. fujikuroi* was obtained from the National Academy Agricultural Science in Korea. Isolate was inoculated in potato dextrose broth (PDB) and cultured at 26 °C under continuous light for one week. The *F. fujikuroi* culture was washed by centrifugation with distilled water, and the spore concentration was adjusted to 1 × 10^6^ spores/mL. Forty seeds per each line were placed in a tissue-embedding cassette (M512, Simport, Beloeil, QC, Canada). Before inoculation, the seeds in the tissue-embedding cassette were surface sterilized with hot water (57 °C) for 13 min, then allowed to drain and cool. Subsequently, the seeds were soaked in the spore suspensions (1 × 10^6^ spores/mL) for 3 d for inoculation with gentle shaking four times a day for equilibration. After inoculation, thirty seeds per line were sown in commercial seedling trays, and seedlings were grown in a greenhouse (28 ± 3 °C day/23 ± 3 °C night, 12 h light). Bakanae disease symptom on each line was evaluated by calculating the proportion of healthy plants at 1 month after sowing. The healthy and non-healthy plants are classified as described by Kim et al. [27]. The plants exhibiting elongation with thin and yellowish-green, stunted growth, and dead seedlings were classified as non-healthy plants.

The plants showing the same phenotype as the untreated plants, slight elongation then normal growth without thin and yellowish-green were regarded as healthy plants.

Statistical differences between means were analyzed using Duncan’s multiple range test after one-way analysis of variance (ANOVA). The level of significance was designated as *p* < 0.05 and was determined using the SAS Enterprise Guide 4.3 program (SAS Institute Inc., Cary, NC, USA).

### 4.3. Localization of F. fujikuroi in Zenith and Ilpum Plants

Shoot bases of 10-day-old seedlings derived from Zenith and Ilpum seeds inoculated by GFP-tagged *F. fujikuroi* isolate CF283 [16] were observed under confocal laser-scanning microscopy (LSM-800, Zeiss, Germany) at GFP channel, and images were obtained using Zeiss LSM Image Browser. All experiments were conducted twice with at least three replicates.

### 4.4. DNA Extraction and Polymerase Chain Reaction

Genomic DNA from young leaf tissue was prepared according to the CTAB method [39] with minor modifications. Polymerase chain reaction (PCR) was performed in 25-µL reaction mixture containing 25 ng template DNA, 10 pmol of each primer, 10× e-Taq reaction buffer, 25 mM MgCl2, 10 mM dNTP mix and 0.02 U of SolGent e-Taq DNA polymerase (SolGent, Daejeon, South Korea). The reaction conditions were set as follows: initial denaturation at 94 °C for 2 min; 35 cycles of denaturation at 94 °C for 20 s, annealing at 57 °C for 40 s and extension at 72 °C for 40 s; and a final extension at 72 °C for 7 min. The amplification products were electrophoresed on a 3% (w/v) agarose gel and visualized by ethidium bromide staining.

### 4.5. QTL Analysis of the F_2:9_ Population and Development of InDel Markers for Fine Mapping

Polymorphic SSR markers (*n* = 164) that were evenly distributed on rice chromosomes were selected from the Gramene database (http://www.gramene.org, accessed on 12 August 2018). These markers were used to construct a linkage map and for QTL analysis of the F_2:9_ populations. The linkage map was constructed using Mapmaker/Exp v.3.0, and the genetic distance was obtained using the Kosambi map function [40]. Putative QTLs were detected using the composite interval mapping (CIM) function in WinQTLcart v.2.5 (WinQTL cartographer software [41]. A logarithm of the odds (LOD) ratio threshold of 3.0 was used to confirm the significance of a putative QTL. InDel markers were developed based on the fragment size differences in the sequence (in the range of 20 bp) between *japonica* (Gramene database http://www.gramene-.org, accessed on 12 August 2018) and *indica* (BGI-RIS; http://rice.genomics.org.cn, accessed on 12 August 2018) in the target region on chromosome 1. The primers were designed using Primer3 software (http://web.-bioneer.co.kr/cgi-bin/primer/primer3.cgi, accessed on 12 August 2018).

## 5. Conclusions

In this study, we identified a new major QTL *qBK1^z^* conferring bakanae disease resistance from a new genetic source of *indica* variety, Zenith. Through QTL analysis and fine mapping, we narrowed down the *qBK1^z^* locus to 730 kb on the short arm of chromosome 1 which is a novel locus compared with all the previously identified bakanae disease resistance QTLs. This new *qBK1^z^* can be introduced to the elite favorable background varieties by a marker-assisted backcrossed breeding together with the previously identified QTLs of the bakanae disease resistance. Furthermore, the information of the localization and different abundance on the vascular bundle, mesophyll tissue and hypodermis of infected stems of infected rice between resistant and susceptible varieties will be useful for further studying the interaction between the pathogen (*F. fujikuroi*) and rice host plants.

## Figures and Tables

**Figure 1 plants-10-00434-f001:**
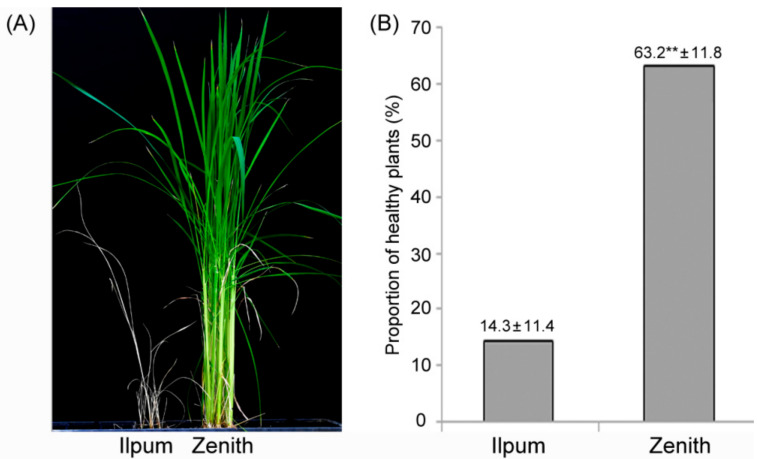
Phenotype (**A**) and proportion of healthy plants (**B**) in Ilpum and Zenith infected with the *Fusarium fujikuroi* isolate CF283. ** Significant at the 1 % level.

**Figure 2 plants-10-00434-f002:**
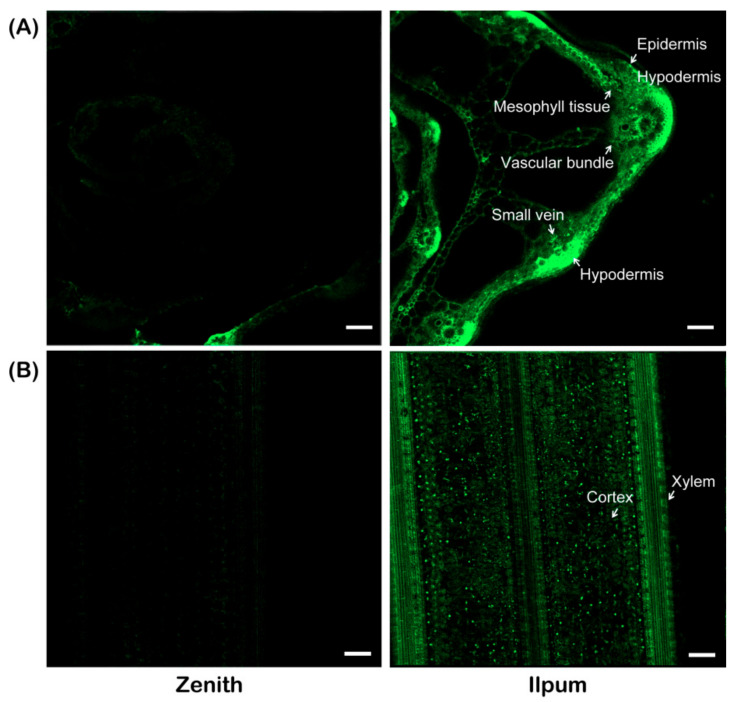
Confocal imaging of Zenith and Ilpum rice plants infected with CF283GFP *Fusarium fujikuroi* isolates. (**A**) Radial and (**B**) longitudinal sections of the basal stem (Scale bar = 50 μm).

**Figure 3 plants-10-00434-f003:**
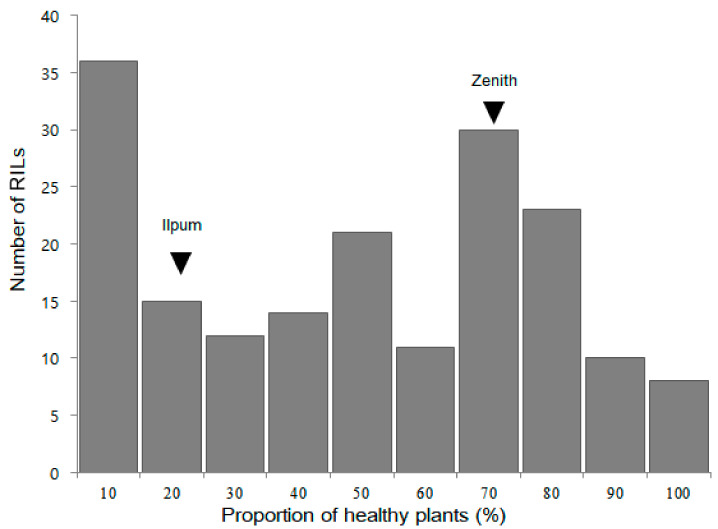
Frequency distribution of the proportion of healthy F_2:9_ plants derived from a cross between Ilpum and Zenith after bakanae disease inoculation. The mean proportion of healthy plants of Ilpum and Zenith are indicated by black arrow heads.

**Figure 4 plants-10-00434-f004:**
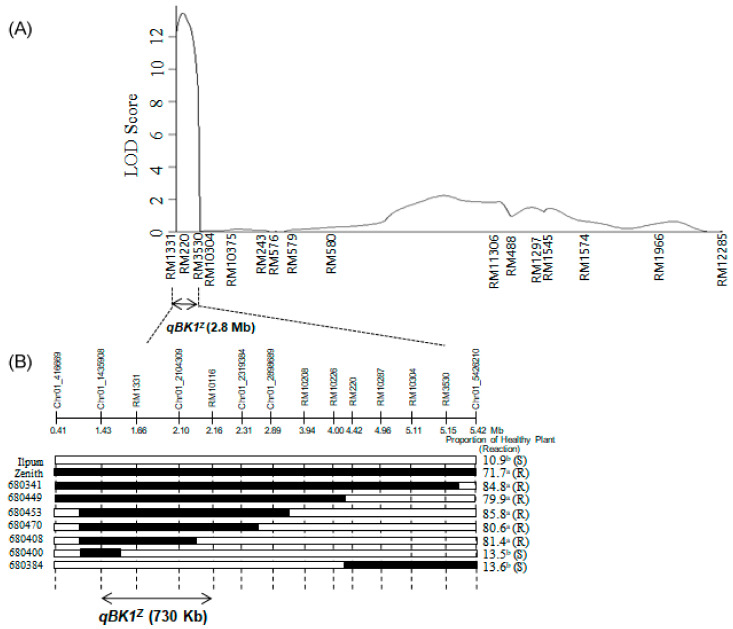
Quantitative trait locus (QTL) analysis of *qBK1^Z^* on chromosome 1 using recombinant inbred lines (RILs) derived from a cross between Ilpum and Zenith. (**A**) In primary mapping, *qBK1^Z^* was founded in the 2.8 Mb region between the RM1331 and RM3530 markers on chromosome 1. (**B**) Location of *qBK1^Z^* was narrowed down to 730 kb region between the markers Chr01_1435908 and RM10116 in secondary mapping using seven selected homozygous recombinants. Black bars show the homozygous regions for Zenith alleles; white bars indicate the homozygous regions for Ilpum alleles; R, resistant to bakanae disease; S, susceptible to bakanae disease. The proportion of healthy plant was calculated from three biological replications. Values (%) of the proportion of healthy plant with different letters are significantly different by Duncan’s multiple range test at *p* = 0.05.

**Figure 5 plants-10-00434-f005:**
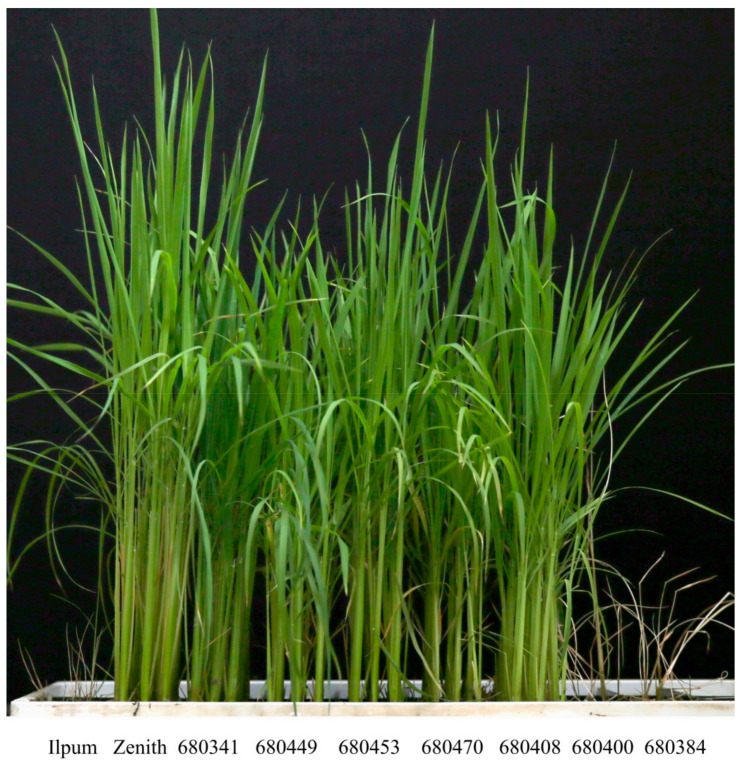
Phenotypic responses to bakanae disease infection in seven homozygous recombinants for secondary mapping.

**Figure 6 plants-10-00434-f006:**
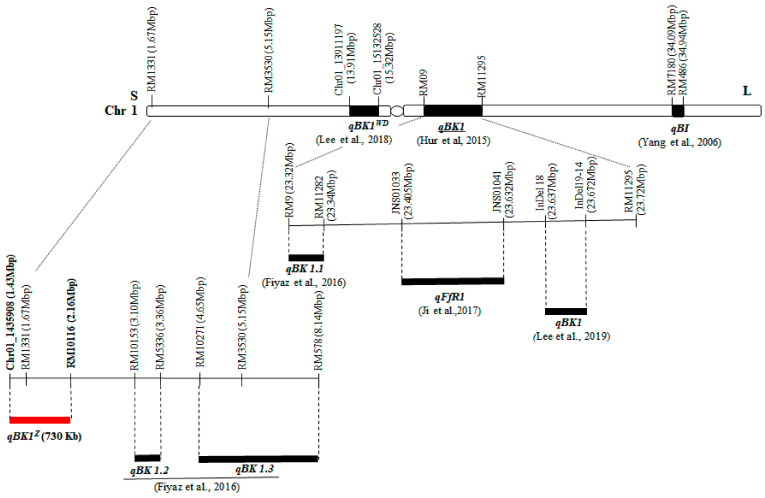
Physical locations of bakanae disease resistance quantitative trait loci on chromosome 1.

**Table 1 plants-10-00434-t001:** Putative quantitative trait locus (QTL) associated with bakanae disease resistance detected at the seedling stage by composite interval mapping of the 180 F_2:9_ populations derived from a cross between Ilpum and Zenith.

QTL	Chromosome	Position (cM)	Marker Interval	LOD ^a^	PVE ^b^ (%)	Additive Effect	Dominant Effect
*qBK1^Z^*	1	4.0	RM1331–RM3530	13.43	30.93	−22.10	−1.09

^a^ LOD: log of odds score; ^b^ PVE: percentage of phenotypic variation explained.

## Data Availability

The data presented in this study are available on request from the corresponding author.

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
