# Peer review of "Mapping of a Major QTL, qBK1Z, for Bakanae Disease Resistance in Rice"

_plants, 2021, doi:10.3390/plants10030434_

Round 1
Reviewer 1 Report
Authors describe fine-scaling of the major QTL for Bakanae disease caused by Giberella fujikuroi.
To answer/adjust eventually:
- How many genes are within the given QTL?
esp. genes linked to gibberellic acid etc, incl. references to other studies
- Why was it possible/main reason to refine the QTL more precisely than the previous studies?
- Any other rice diseases with QTLs in proximity?
- Why less crop loss in different continents?
- Refs. to other studies analysing Bakanae with different genotypes available, if so compare and cite
- Is this refined QTL in proximity to other syntenic regions in other crops (such as mays, sorghum, etc.?)
- In "Management of Bakanae disease of rice" noticed that specific fungicides could work for Bakanae,
in Introduction you noted, that these would not work?
Fig 3. Maybe quality of the figure can be improved using more advanced visualisation ggplot2
Fig 4. Maybe resolution can be improved.
Fig 5. Are the genotypes '680341' described somewhere?
Author Response
Response to Reviewer 1 Comments
Point 1: How many genes are within the given QTL? esp. genes linked to gibberellic acid etc, incl. references to other studies
Response 1: Thank you for your comment. The position of the qBK1z was mapped to an approximate 730 kb interval in the physical map between Chr01_1435908 (1.43 Mbp) and RM10116 (2.16 Mbp) markers. According to RAP-DB (http://rice.plantbiology.msu.edu/index.shtml), there are 108 genes between the two markers in the reference genome of rice (Nipponbare). We think that current region of qBK1z is enough to utilize in marker-assisted selection, but it is too wide to address the number of the genes and some potential candidate genes. We will provide more detailed information including the function of the genes within the region after narrowing down the position of this QTL in the further study.
Point 2: Why was it possible/main reason to refine the QTL more precisely than the previous studies?
Response 2: Recently, many QTL studies are conducted using several resistant varieties and some of them has narrower intervals than that of the qBK1z (730 kb). Identifying new resistance genes from diverse sources is important for rice breeding programs to acquire durable resistance against bakanae disease by enhancing the resistance level and/or help to overcome the breakdown of resistance genes. This is the first report to identify a major QTL of bakanae disease resistance from an indica variety Zenith. We aimed to provide a new genetic source, qBK1z with detailed gene locus information for developing resistant rice lines which contains single or multiple major QTLs to enhance bakanae disease resistance.
Point 3: Any other rice diseases with QTLs in proximity?
Response 3: As advised by reviewer 1, we searched the disease related QTLs/genes which are located at the qBK1z locus between Chr01_1435908 (1.43 Mbp) and RM10116 (2.16 Mbp). There was no disease resistant QTLs/genes within the current region of qBK1z, and the nearest resistant gene was blast resistant Pit gene on 2.68 Mb. But this gene was still far from qBK1z to add in the manuscript.
Point 4: Why less crop loss in different continents?
Response 4: Thank you for your comment. Bakanae disease was also reported in the other continent such as USA, Italia etc. We could not find a report on yield loss. One of the reason bakanae diseases is less severe in the America or European countries might direct seeding is more prevail in those countries. With the Reviewer’s advice, authors revised the manuscript and a reference [15] as below.
→ (Page 2) Germinating rice seeds in seed boxes for mechanical transplantation has caused many problems associated with diseases [15] including bakanae disease, which are not considered serious in direct seeding.
Point 5: Refs. to other studies analyzing Bakanae with different genotypes available, if so compare and cite
Response 5: We described the previously identified QTLs associated with bakanae disease from various genotypes in the Introduction (please see page 2). We also addressed the positional relationship between qBK1z and the QTLs about bakanae disease resistant on chromosome 1 in the Discussion section (please see Page 7 and Figure 6).
Point 6: Is this refined QTL in proximity to other syntenic regions in other crops (such as mays, sorghum, etc.?)
Response 6: Thank you for your comment. We think that the current position of the qBK1z was still large to analyze the Syntenic region with other species. We will provide more detailed information including the function of the genes within the region after narrowing down the position of this QTL in the further study.
Point 7: In "Management of Bakanae disease of rice" noticed that specific fungicides could work for Bakanae, in Introduction you noted, that these would not work?
Response 7: Application of fungicide is one of the most common ways to control bakanae disease in rice cultivation. As we described in the Introduction, application of fungicide are insufficient. The application of fungicides is not functioning well for destroying the spores of this fungal pathogen, and some pathogen showed resistance to the fungicides. Therefore, the genetic improvement of rice using the QTLs/genes providing the bakanae disease resistance would be a more effective way to control bakanae disease.
Point 8: Fig 3. Maybe quality of the figure can be improved using more advanced visualisation ggplot2
Response 8: Thank you for the comment. Based on the author’s guidelines, we incorporated all the figures in word file for the initial submission. Once our manuscript is accepted, we will send the original high resolution files for all figures to the Plants editorial office.
Point 9: Fig 4. Maybe resolution can be improved.
Response 9: Thank you for your comment. Once current manuscript is accepted, we will send the original files about all figures to the Plants editorial office.
Point 10: Fig 5. Are the genotypes '680341' described somewhere?
Response 10: We have not been described the genotypes '680341'.

Reviewer 2 Report
This work identified and mapped a novel QTL, qBK1z, which gives rice plants resistance against bakanae disease, using 180 RILs between resistant line Zenith and susceptible line Ilpum. Besides, authors sorted several QTL loci as well as qBK1z on Chr.1. Although this work is simple, as authors mentioned, pyramiding usage of multiple QTL loci becomes more effective against the disease in rice than single. Here, I raised a few points to be addressed before publication as below.
Authors narrowed down qBK1z locus within 730 kb of Chr.1. How many genes are in this region? What type of genes among them are considered as candidates conferring resistance? The information should be described in the manuscript. Besides, presentation of their expression data might be helpful for upgrading this work.
Line 148. “qSTV11z” may be wrong. Please check it.
Line 166. It is kind to add “(Fig. 2)” behind “… isolate CF283”.
Line 197. Delete “In”.
Author Response
Response to Reviewer 2 Comments
This work identified and mapped a novel QTL, qBK1z, which gives rice plants resistance against bakanae disease, using 180 RILs between resistant line Zenith and susceptible line Ilpum. Besides, authors sorted several QTL loci as well as qBK1z on Chr.1. Although this work is simple, as authors mentioned, pyramiding usage of multiple QTL loci becomes more effective against the disease in rice than single. Here, I raised a few points to be addressed before publication as below.
Point 1: Authors narrowed down qBK1z locus within 730 kb of Chr.1. How many genes are in this region? What type of genes among them are considered as candidates conferring resistance? The information should be described in the manuscript. Besides, presentation of their expression data might be helpful for upgrading this work.
Response 1: Thank you for your comment. The position of the qBK1z was mapped to an approximate 730 kb interval in the physical map between Chr01_1435908 (1.43 Mbp) and RM10116 (2.16 Mbp) markers. According to RAP-DB (http://rice.plantbiology.msu.edu/index.shtml), there are 108 genes between the two markers in the reference genome of rice (Nipponbare). We think that current region of qBK1z is enough to utilize in marker-assisted selection, but it is too wide to address the number of the genes and some potential candidate genes. Authors think that we can provide more detailed information including the information about the candidate genes within the region and their expression patterns after narrowing down the position of this QTL in the further study.
Point 2: Line 148. “qSTV11z” may be wrong. Please check it.
Response 2: Thank you very much. It was our mistake. We corrected it to qBK1z.
Point 3: Line 166. It is kind to add “(Fig. 2)” behind “… isolate CF283”.
Response 3: Thank you very much. We added ‘(Fig. 2)’ as reviewer advised.
Point 4: Line 197. Delete “In”.
Response 4: Thank you for pointing out our mistake. We deleted ‘In’.

Round 2
Reviewer 1 Report
all remarks were properly addressed
Author Response
Response to Comments
We addressed about the all comments from the editor.
Authors is going to send the all figures (include revised ones) to the Ms. Effy Xia (Assistant Editor) by email.